# Differentiation and Interconnection of the Bacterial Community Associated with *Silene nigrescens* Along the Soil-To-Plant Continuum in the Sub-Nival Belt of the Qiangyong Glacier

**DOI:** 10.3390/plants14081190

**Published:** 2025-04-11

**Authors:** Wangchen Sonam, Yongqin Liu, Luming Ren

**Affiliations:** 1State Key Laboratory of Tibetan Plateau Earth System, Environment and Resources (TPESER), Institute of Tibetan Plateau Research, Chinese Academy of Sciences, Beijing 100101, China; 2University of Chinese Academy of Sciences, Beijing 100049, China; 3Center for the Pan-Third Pole Environment, Lanzhou University, Lanzhou 730000, China; 4Nanning Garden Expo Park Management Center, Nanning Institute of Tropical Botany, Nanning 530299, China; wangqingsainao@163.com

**Keywords:** differentiation, interconnection, bacteria community, *Silene nigrescens*, sub-nival belt

## Abstract

Plant microbiomes provide significant fitness advantages to their plant hosts, especially in the sub-nival belt. Studies to date have primarily focused on belowground communities in this region. Here, we utilized high-throughput DNA sequencing to quantify bacterial communities in the rhizosphere soil as well as in the root and leaf endosphere compartments of *Silene nigrescens* to uncover the differentiation and interconnections of these bacterial communities along the soil-to-plant continuum. Our findings reveal that the bacterial communities exhibit notable variation across different plant compartment niches: the rhizosphere soil, root endosphere, and leaf endosphere. There was a progressive decline in diversity, network complexity, network modularity, and niche breadth from the rhizosphere soil to the root endosphere, and further to the leaf endosphere. Conversely, both the host plant selection effect and the stability of these communities showed an increasing trend. Total nitrogen and total potassium emerged as crucial factors accounting for the observed differences in diversity and composition, respectively. Additionally, 3.6% of the total amplicon sequence variants (ASVs) were shared across the rhizosphere soil, root endosphere, and leaf endosphere. Source-tracking analysis further revealed bacterial community migration among these compartments. The genera *Pseudomonas*, *IMCC26256*, *Mycobacterium*, *Phyllobacterium*, and *Sphingomonas* constituted the core of the bacterial microbiome. These taxa are shared across all three compartment niches and function as key connector species. Notably, *Pseudomonas* stands out as the predominant taxon among these bacteria, with nitrogen being the most significant factor influencing its relative abundance. These findings deepen our understanding of the assembly principles and ecological dynamics of the plant microbiome in the sub-nival belt, offering an integrated framework for its study.

## 1. Introduction

The sub-nival belt, situated near the alpine snowline, represents the highest terrestrial habitat on Earth [1]. In this extreme environment, plants operate at the brink of their physiological and ecological limits [1,2,3]. However, these plants are not isolated entities; they function as holobionts, consisting of the host plants and their associated microbiomes [4]. The microbiome, which inhabits the rhizosphere, roots, and other plant tissues, plays a crucial role in enhancing plant growth, health, and resilience to environmental stressors [5,6,7]. Given the importance of these microbial partnerships, it is imperative to deepen our understanding of plant microbiomes in sub-nival ecosystems to uncover how plants survive and adapt to climate change in such extreme environments.

Different plant compartment niches, such as the rhizosphere soil, root endosphere, and leaf endosphere, provide unique biotic and abiotic conditions that support diverse microbial communities [8,9,10]. The microbial communities in these niches vary significantly in density, diversity, and composition [5,11,12]. Moreover, these communities display unique life strategies based on the plant compartment they occupy, with their responses to environmental conditions being specific to these niches [13,14,15,16]. Despite these differences, previous studies have suggested potential interconnections between microbiomes in different plant compartment niches [10,17,18]. Our previous research has also shown that numerous microbial taxa are shared across different plant compartment niches, indicating the existence of potential pathways for microbial exchange among these diverse niches [19,20]. These findings underscore the need for an integrated, systematic approach to studying plant–microbe interactions, which is particularly important given the differing phenological responses of aboveground and belowground plant components to climate change [21]. However, research on the plant microbiome in the sub-nival belt has predominantly focused on belowground microbial communities, such as those in the rhizosphere and root-associating microbes [22,23,24,25,26,27,28,29]; there has been limited exploration into the differentiation and interconnections of microbial communities along the soil-to-plant continuum in these regions.

In the process of coevolution, plants attract beneficial microbes through specific signals and use their immune systems and nutrients to select them [30,31,32]. Additionally, a growing body of evidence suggests that microbiome assembly along the soil–plant continuum is primarily determined by the plant compartment niche, rather than environmental factors such as site location or fertilization practices [15,18,33,34,35,36,37]. These studies underline the crucial role of host plant identity or plant compartment niche in shaping microbial communities. However, it is important to note that the levels and ratios of nitrogen, phosphorus, and potassium—the three essential nutrients for plant growth and development—significantly influence plant health [38,39,40]. These nutrients directly or indirectly affect the formation, stabilization, and maintenance of plant-associated microbial diversity [41,42,43,44]. Therefore, exploring the relationship between the levels and ratios of these essential nutrients in plants and their associated microbial communities is crucial. However, our understanding of how these nutrient dynamics affect microbial communities associated with plants in the sub-nival belt remains limited.

The genus *Silene* (Caryophyllaceae) exhibits remarkable diversification in the Tibetan Plateau [45]. Among its diverse species, *S. nigrescens* emerges as an endemic species uniquely adapted to the sub-nival belt [46]. Its perennial growth habit and sustained nutrient acquisition requirements establish this species as an ideal model system for studying both early successional dynamics and climate-mediated resilience mechanisms. In this study, we employed next-generation high-throughput sequencing to analyze bacterial communities within distinct compartment niches of *S. nigrescens*, including the rhizosphere soil, root endosphere, and leaf endosphere. Additionally, we quantified total nitrogen (TN), phosphorus (TP), and potassium (TK) in the soil, roots, and leaves. We aimed to answer the following questions: (i) How do bacterial communities associated with *S. nigrescens* vary across different plant compartment niches? Furthermore, how are TN, TP, and TK levels and their ratios correlated with variations in bacterial diversity and community composition? (ii) How are bacterial communities in different plant compartment niches interrelated, and is there a core taxon that links these communities? By addressing these questions, this study will not only advance our understanding of the assembly principles and ecological interactions of the plant microbiome in the sub-nival belt but also provide an integrated perspective on it.

## 2. Results

### 2.1. Variation in Community Diversity and Composition Across Plant Compartment Niches, and Their Correlation with the Levels and Ratios of TN, TP, and TK

#### 2.1.1. Variation in Community Diversity Across Plant Compartment Niches

High-throughput Illumina sequencing generated a total of 406,306 raw reads derived from 30 microbial DNA libraries, each corresponding to a unique sample collected along the soil–plant continuum. After quality control, chimera removal, and denoising steps, 249,642 high-quality reads were retained. These high-quality reads were clustered into 1576 ASVs. To ensure a standardized sampling depth for downstream analyses, the number of reads per sample (ranging from 6368 to 22,395) was rarefied to the minimum read count of 6368. This rarefaction process resulted in a final dataset consisting of 1551 bacterial ASVs. Based on our analyses, the rarefaction curves for ASV richness and Shannon diversity reached a saturation plateau (Appendix A), indicating that the majority of the diversity present in the plant microbiome was effectively captured.

The alpha diversity indices, including Richness, Shannon, Pielou, and ACE, were significantly affected by the plant compartment niche (*p* < 0.01) (Figure 1a–d). Among the plant compartment niches, the rhizosphere soil exhibited the highest alpha diversity, followed by the root endosphere, whereas the leaf endosphere showed the lowest diversity (Figure 1a–d).

#### 2.1.2. Variation in Community Composition Across Plant Compartment Niches

There was significant differentiation in community composition across different plant compartment niches (Table 1). At the genus level, variations in the relative abundance of the top 10 taxa were observed among these compartment niches. *Pseudomonas*, with a relative abundance of 93.28%, accounted for the greater relative abundance of taxa in the leaf endosphere (Figure 2a). In the root endosphere, a higher relative abundance of taxa was primarily attributed to *Pseudomonas* (55.53%), followed by *Conyzicola* (12.09%), *Devosia* (1.73%), *Allorhizobium*-*Neorhizobium*-*Pararhizobium*-*Rhizobium* (1.72%), *Pseudonocardia* (1.17%), and *Actinoplanes* (1.60%) (Figure 2a). By contrast, in the rhizosphere soils, taxa with higher relative abundance included *Pseudomonas* (29.25%), *Blastococcus* (5.01%), *Nocardioides* (3.89%), *Pseudonocardia* (2.49%), *Mycobacterium* (2.46%), and *Sphingomonas* (2.38%) (Figure 2a).

Indicator species analysis revealed that there were five indicator ASVs for the leaf endosphere, with four of these ASVs belonging to members of the *Pseudomonas* genus (Table 2). For the root endosphere, there were four indicator ASVs, belonging to the genera *Conyzicola*, *Polaromonas*, *Brevundimonas*, and *Delftia* (Table 2). By contrast, the rhizosphere soil had 15 indicator ASVs. Of these, four were from *Blastococcus*, two from *Nocardioides*, two from *Allorhizobium*-*Neorhizobium*-*Pararhizobium*-*Rhizobium*, and the remaining ASVs were associated with members of the Frankiales, *Gaiella*, *MB-A2-108*, Microbacteriaceae, Beijerinckiaceae, Sporichthyaceae, and Gemmatimonadaceae (Table 2).

The plant compartment niche/bacteria preference analysis demonstrated that specific bacterial ASVs were favored by the three plant compartment niches, with a strong preference observed in the leaf endosphere (*p* < 0.0001) (Figure 3a). Additionally, 13 abundant bacterial ASVs exhibited significant preferences for plant compartment niches (*p* < 0.001). These include ASV1–3 and 5–7 (*Pseudomonas*), ASV11 (*Actinoplanes*), ASV12 (*Carnobacterium*), ASV18 (*Bosea*), ASV32 (*Delftia*), ASV33 (*Allorhizobium*-*Neorhizobium*-*Pararhizobium*-*Rhizobium*), ASV48 *(Blastococcus*), and ASV51 (Beijerinckiaceae). In our dataset, six pairs of plant compartment niches and abundant bacterial ASVs exhibited remarkably strong preferences, with two-dimensional preference (2DP) values greater than 3. These pairs include the leaf endosphere with ASV7 (*Pseudomonas*) and ASV12 (*Carnobacterium*), and the rhizosphere soil with ASV48 (*Blastococcus*), ASV51 (Beijerinckiaceae), and ASV54 (Sporichthyaceae) (Figure 3c,d).

#### 2.1.3. Correlation Between TN, TP, and TK Levels and Ratios and Variations in Community Diversity and Composition

The optimized aggregated boosted tree model identified total nitrogen (TN) as the primary determinant influencing variations in alpha diversity indices, including Richness (53.50%), Shannon (48.94%), Pielou (40.43%), and ACE (61.43%) (Figure 1e–h).

Mantel test results revealed that TK (Mantel test, *r* = 0.692, *p* = 0.0001) and TN (*r* = 0.599, *p* = 0.0004) were the most significant factors influencing community compositions. At the genus level, the optimized aggregated boosted tree model further indicated that TN was the most significant factor in explaining variations in the relative abundance of several bacterial genera: *Pseudomonas* (66.34%), *Nocardioides* (77.99%), *Sphingomonas* (47.51%), *Allorhizobium*-*Neorhizobium*-*Pararhizobium*-*Rhizobium* (38.09%), and *Devosia* (76.10%) (Figure 2b,e,g,h,k). The ratio of TN to TP accounted for 41.55% of the variation in the relative abundance of *Conyzicola* (Figure 2c). Meanwhile, the TP to TK ratio explained 64.51% and 38.11% of the variation in the relative abundance of *Blastococcus* and *Actinoplanes*, respectively (Figure 2d,i). Additionally, TP was responsible for 47.75% and 31.87% of the variation in the relative abundance of *Pseudonocardia* and *Mycobacterium*, respectively (Figure 2f,j).

### 2.2. Variation in Community Co-Occurrence Networks Across Plant Compartment Niches

The construction of correlation-based networks of the bacterial communities resulted in three networks, consisting of 35, 152, and 359 nodes connected by 66, 932, and 3424 edges, respectively (Table 3). The values of the average clustering coefficient, average path length, and modularity of each plant compartment niche Erdős-Rényi random network were lower than those of the respective observed network, suggesting that the observed network was a non-random and modular structure (Table 3).

We found more modular structures in rhizosphere soil and root endosphere networks due to their larger modularity values (Table 3). Similarly, the node degree was significantly larger for the rhizosphere soil and root endosphere networks than for the leaf endosphere network (Figure 4b). In contrast, the node closeness centrality within both the rhizosphere soil network and the root endosphere network was significantly lower compared to that of the leaf endosphere networks (Table 3). Additionally, we identified 66, 26, and 4 nodes as connectors in the rhizosphere soil, root endosphere, and leaf endosphere networks, respectively (Appendix A). The species composition of connectors varied significantly across different plant compartments. For instance, all connectors in the leaf endosphere network were identified as *Pseudomonas* (family Pseudomonadaceae). In the rhizosphere soil network, most connectors belonged to families such as Geodermatophilaceae (e.g., *Geodermatophilus*, *Blastococcus*, *Modestobacter*), Pseudonocardiaceae (e.g., *Lechevalieria*, *Pseudonocardia*), Solirubrobacteraceae (e.g., *Solirubrobacter*, *Conexibacter*), Microbacteriaceae (e.g., *Microbacterium*, *Clavibacter*, *Rathayibacter*), Beijerinckiaceae (e.g., *Methylobacterium-Methylorubrum*, *Microvirga*), Devosiaceae (e.g., *Devosia*), Pseudomonadaceae (e.g., *Pseudomonas*), and Sphingomonadaceae (e.g., *Sphingomonas*) (Appendix A).

### 2.3. Variation in Community Stability and Host Selection Pressure Across Plant Compartment Niches

The niche breadth and AVD of the community showed a gradual decline as they transitioned from the rhizosphere soil to the root endosphere, and further to the leaf endosphere (Figure 5a,b). Furthermore, we assessed the depleting effect of host plants on their associated bacterial taxa. Host plants exerted significant selection pressure on taxa originating from the rhizosphere soil (Figure 5c,d). For instance, stronger selection pressure was observed on taxa within the leaf endosphere, as indicated by the highest DI value (14.07) recorded in this region (Figure 5d).

### 2.4. Interconnections Among Communities Across Plant Compartment Niches and Identification of Core Taxa

#### 2.4.1. Interconnections Among Communities Across Plant Compartment Niches

The 56 ASVs (3.6% of the total ASVs) shared across the rhizosphere soil, root endosphere, and leaf endosphere predominantly belong to the genera *Pseudomonas*, *Pseudarthrobacter*, *Nocardioides*, *IMCC26256*, *Gaiella*, *Bradyrhizobium*, and *Sphingomonas* (Figure 6a, Appendix A). Meanwhile, 222 ASVs (8.6% of the total ASVs) that are common to both the rhizosphere soil and root endosphere mainly originate from the genera *Pseudomonas*, *Pseudonocardia*, *Allorhizobium-Neorhizobium-Pararhizobium-Rhizobium*, *Nocardioides*, *Mycobacterium*, *Bradyrhizobium*, *Blastococcus*, *Aureimonas*, *Solirubrobacter*, and *Sphingomonas* (Figure 6a, Appendix A). Similarly, 100 ASVs (6.4% of the total ASVs) shared between the rhizosphere soil and leaf endosphere are primarily associated with the genera *Pseudomonas*, *Sphingomonas*, *IMCC26256*, *Nocardioides*, *Gaiella*, *Conyzicola*, *Brevundimonas*, and *Arthrobacter* (Figure 6a, Appendix A). Additionally, the 103 ASVs (6.6% of the total ASVs) shared between the root endosphere and leaf endosphere predominantly correspond to the genera *Pseudomonas*, *Nocardioides*, *Massilia*, *IMCC26256*, *Corynebacterium*, *Brevundimonas*, *Chryseobacterium*, and *Bradyrhizobium* (Figure 6a, Appendix A).

Source-tracking analysis indicated that 6.14% of the taxa in the root endosphere originated from rhizosphere soil, while 50.93% was derived from the leaf endosphere (Figure 6b). By contrast, 91.51% of the taxa in the leaf endosphere were traced back to the root endosphere (90.34%) and rhizosphere soil (1.17%) (Figure 6b). Furthermore, we found that the *Pseudomonas* and *Delftia* were significantly enriched and overlapped in two plant compartment niches (root endosphere and leaf endosphere) (Figure 6c). However, taxa from *Conexibacter*, *Blastococcus*, *67-14*, and *Nakamurella* were significantly depleted in the endospheres of both leaves and roots (Figure 6d).

#### 2.4.2. Core Taxa Identification

A Zi-Pi plot was generated to analyze the topological roles of nodes in the co-occurrence network, encompassing all taxa present in the rhizosphere soil, root endosphere, and leaf endosphere (Figure 6e,f, Appendix A). The analysis revealed that 52 node ASVs (18.64% of all node ASVs) were identified as connectors (Appendix A), while the remaining 227 node ASVs (81.72%) were classified as peripherals. Notably, among these connectors, eight node ASVs were consistently found in the rhizosphere soil, root endosphere, and leaf endosphere, and were identified as core taxa. These core taxa belong to the following genera: *Pseudomonas* (accounting for 50% of all core ASVs), *IMCC26256*, *Mycobacterium*, *Phyllobacterium*, and *Sphingomonas* (Appendix A).

## 3. Discussion

### 3.1. Variations in Community Across Plant Compartment Niches

Plants provide varied compartment niches for microbes, each with distinct environmental conditions. The rhizosphere is a complex, nutrient-rich environment supported by root-exuded carbon compounds and nutrients [47]. The root system, by contrast, offers an internal environment with high humidity and stable nutrient availability, while leaves are subject to external factors such as light, temperature, and atmospheric composition [48]. These differing conditions lead to specific microbial community structures. Rhizospheric microbes benefit from root-released nutrients, whereas plants selectively curate microbes in the root and leaf endospheres based on nutrient availability and immune defenses [48,49]. Our study shows that the diversity, composition, niche breadth, stability, and co-occurrence network properties of bacterial communities vary significantly across the rhizosphere soil, root endosphere, and leaf endosphere. Notably, the diversity, network complexity, network modularity, and niche breadth progressively decrease from the rhizosphere to the root endosphere and then to the leaf endosphere. These differences are largely driven by the unique conditions and selective pressures of each plant compartment niche. Additionally, these results further support the view that the rhizosphere provides diverse and suitable ecological niches for microbial communities [50]. In contrast, the internal plant environments are more selective, which limits the number of microbial species that can successfully adapt and persist [51].

The stability of a microbiome is primarily attributed to species diversity, with a general consensus that biodiversity positively influences microbiome stability [52]. For example, previous studies have demonstrated that higher diversity can enhance community stability [53], supporting the notion that diverse ecosystems are better equipped to adapt to and withstand environmental disturbances. However, in this study, we found a negative correlation between the diversity and stability of bacterial communities (Appendix A). Notably, we observed that the leaf endosphere, despite having the lowest bacterial diversity, exhibited the highest community stability (Figure 1a–d and Figure 5b). This apparent paradox can be explained by its co-occurrence network, which is characterized by high closeness centrality and low modularity. High closeness centrality indicates well-connected nodes within the network, enabling the rapid flow of information and resources across the community [54]. This connectivity boosts collective responsiveness to environmental changes. On the other hand, low modularity points to a more integrated network structure, with fewer distinct modules and increased interactions, contributing to overall stability [55].

We identified a significant difference in the indicator taxa and connector taxa among the rhizosphere soil, root endosphere, and leaf endosphere. This indicates that bacterial communities exhibit a high degree of compartmentalization and specialization, reflecting distinct ecological dynamics and functional roles within each plant compartment niche. This adaptation may involve physiological and metabolic traits that confer advantages under the particular conditions of each compartment niche, such as nutrient availability, host interactions, and defense strategies [4]. Additionally, this study revealed a significant preference between specific taxa and plant compartment niches, suggesting a dynamic evolutionary relationship from a coevolutionary perspective. This reciprocal adaptive process underscores the complexity and interdependence within plant–microbe ecosystems, highlighting the importance of considering evolutionary history in studies of plant–microbe interactions.

In the rhizosphere soil, taxa such as *Blastococcus*, Beijerinckiaceae, and Sporichthyaceae function as indicator species, showing a strong preference for this environment. This suggests they are well adapted to the unique conditions of the rhizosphere, indicating a symbiotic relationship where both the microorganisms and the rhizosphere benefit, potentially through nutrient exchange and enhanced root health [47]. Such mutual preferences highlight the adaptive strategies these taxa employ to thrive in their specific niche. In the root endosphere, *Conyzicola*, *Polaromonas*, *Brevundimonas*, and *Delftia* were identified as indicator species, with *Conyzicola* and *Delftia* also acting as connector genera. The dual role of *Conyzicola* and *Delftia* as indicators and connectors highlights their importance in enhancing nutrient dynamics and supporting a stable microbial community in the root endosphere, which was supported by previous studies. For instance, research on maize endophytes has shown that *Conyzicola* can solubilize insoluble zinc compounds and produce siderophores [56]. Additionally, *Delftia* species have been found to produce indole-3-acetic acid, fix atmospheric nitrogen, and solubilize essential minerals such as phosphorus and potassium [57]. In the leaf endosphere, the bacterial microbiome was primarily composed of *Pseudomonas*, which acted as both an indicator and connector genus for leaf endophytes. Moreover, there was a significant mutual preference between *Pseudomonas* and leaf endosphere. The dominance of *Pseudomonas* in the leaf endosphere, serving as both an indicator and connector genus, is largely due to its exceptional metabolic diversity and ability to form beneficial interactions with host plants, such as enhancing nutrient uptake or providing pathogen resistance [58]. These abilities give *Pseudomonas* a significant adaptive advantage, allowing it to effectively colonize and thrive within the leaf endosphere, which in turn supports its role as a key component of the microbial community. This mutual preference likely results from coevolutionary dynamics that optimize plant–microbe interactions for mutual benefit.

### 3.2. Correlation Between the Levels and Ratios of TN, TP, and TK and Community Variations

The concentrations and ratios of nitrogen, phosphorus, and potassium influence plant-associated microbial communities both directly and indirectly. For instance, an increase in nitrogen levels can stimulate the proliferation of bacteria that utilize nitrogen, thereby enhancing nitrogen availability to plants [59]. In contrast, imbalanced nutrient ratios may inhibit beneficial microbes such as mycorrhizal fungi, which impairs the efficiency of phosphorus uptake [60]. Additionally, adequate potassium levels can support the presence of bacteria that contribute to disease resistance, fostering a healthier and more resilient microbial community within the rhizosphere [61,62].

Our findings indicate that total nitrogen was the crucial factor in accounting for the differences in diversity and composition. This strong correlation underscores the critical role of nitrogen in shaping the diversity of bacterial communities associated with *S. nigrescens*, likely due to its pivotal functions in metabolic processes and overall productivity of these ecosystems. The ratios of total nitrogen, total phosphorus, and total potassium have significantly impacted the diversity (e.g., Shannon and Chao1 indices) and composition of the bacterial community associated with *S. nigrescens*. These findings underscore the importance of considering the combined effects of nitrogen, phosphorus, and potassium, as their integrative influence significantly impacts microbial diversity and composition. This comprehensive interaction likely reflects complex nutrient cycling processes and their synergistic effects on the structure of microbial communities associated with *S. nigrescens*. Additionally, our findings indicate that total potassium is the most significant factor influencing bacterial community composition. This supports previous research emphasizing potassium as a crucial element in various biochemical pathways within both plants and microbes. For example, potassium is vital for enzyme activation and osmoregulation, influencing plant growth and microbial metabolism [61,63].

### 3.3. Interconnections of Communities Across Plant Compartment Niches

The results of our current study reveal that numerous ASVs are shared among the rhizosphere soil, root endosphere, and leaf endosphere. We further assess the source and sink relationships of the bacterial communities associated with the rhizosphere soil, root endosphere, and leaf endosphere. The results indicate that the rhizosphere soil contributes 6.14% to the root endophytes and 1.17% to the leaf endophytes, respectively. Moreover, root and leaf endophytes serve as sources and sinks for one another. On the one hand, these results indicate an interconnection of bacterial taxa among the rhizosphere soil, root endosphere, and leaf endosphere; on the other hand, these results demonstrate that the nearby species pool is a potential source of plant endophytes. In line with our results, similar findings have also been reported in crop microbiomes [10,18]. However, contrary to previous findings suggesting that soils serve as a primary reservoir for plant-associated microbiomes [8,64,65], our results reveal a surprisingly low contribution from the rhizosphere soil, with only 6.14% of the bacteria in the root endosphere originating from this source. Previous studies have shown that the adaptive strategies of *Silene acaulis* in the sub-nival belt are primarily attributed to the morphology and functions of its canopy, rather than rhizosphere-mediated effects that recruit and foster growth-promoting microbial communities [26]. This characteristic of *Silene acaulis* may potentially disrupt the interconnection between rhizosphere microbes and endophytes. Furthermore, seeds serve as a primary vehicle for parent plants to transmit microbes to their offspring [66]. Consequently, we hypothesize that seed-borne bacteria are likely to be the predominant source of bacteria associated with *Silene* species.

In this study, we identified bacterial taxa (*Pseudomonas*, IMCC26256, *Mycobacterium*, *Phyllobacterium*, and *Sphingomonas*) as universally present across the rhizosphere, root endosphere, and leaf endosphere of *S. nigrescens*. These taxa, along with *Conexibacter*, *Arthrobacter*, *Aureimonas*, *Devosia*, *Pseudarthrobacter*, and *Pseudonocardia*, have been characterized as key connector species in shaping bacterial community networks. Notably, *Pseudomonas* was the predominant genus in the rhizosphere, root, and leaf endospheres, with significant enrichment in the leaf and root endospheres. These findings demonstrate that these bacterial taxa, particularly those within the *Pseudomonas* genus, form the core microbiome of *S. nigrescens*, playing a crucial role in maintaining the integrity and stability of its associated bacterial community. Studies have shown that taxa within the *Pseudomonas* genus possess a remarkable ability to colonize plant environments, promote plant growth, and inhibit plant pathogens [58]. The *Mycobacterium* is known for its flexibility in energy metabolism, and this taxon can significantly enhance the stress tolerance of host plants through 1-aminocyclopropane-1-carboxylic acid deaminase activity [67,68]. *Arthrobacter* has been reported as part of the core microbiota for many plants [69,70,71]. The genome of *Arthrobacter* sp. contains several genes responsible for the synthesis of plant hormones, and its inoculation has demonstrated a significantly positive effect on plant growth [72]. *Sphingomonas* are usually found in multiple parts (roots, leaves, flowers) of mature plants and can promote plant growth and enhance plant resistance to osmotic stress [73]. *Conexibacter* has been shown to enhance the stability of microbial communities in adverse environmental conditions [74]. Genome analysis of taxa within *Aureimonas* revealed traits for exopolysaccharide synthesis, protein secretion, biofilm production, and stress adaptation, as well as a significant positive impact on plant health [75]. *Devosia* is a symbiotic nitrogen-fixing bacterium that can enhance plant growth through the production of siderophores and indole-3-acetic acid [76,77]. The taxa belonging to *Pseudarthrobacter* are potential cold-adapted bacteria [78] and have shown beneficial effects on plant growth and flavonoid content [79]. Strains of the *Pseudonocardia* genus, isolated from plant tissues like stems, leaves, and root nodules [80], are of interest for their ability to produce glycosylated polyenes and novel antifungal compounds [81]. Based on the evidence from these studies, we summarize the characteristics of the core bacteria of *S. nigrescens* as follows: (1) they can colonize various plant environments, (2) they possess beneficial traits for the plant, (3) they play a crucial role in maintaining the stability and integrity of the microbial community, and (4) *Pseudomonas* is the notably predominant taxon. Furthermore, the theoretical framework of coevolution suggests that plants and their microbiomes have evolved together over millions of years, with most of these interactions being mutually beneficial [30,32]. Consequently, we speculate that the symbiotic relationship between *S. nigrescens* and its core bacteria may represent a key adaptive mechanism that enables *S. nigrescens* to withstand the harsh conditions of the sub-nival belts.

Previous studies have demonstrated that certain taxa within the *Pseudomonas* genus possess nitrogen-fixing capabilities [82]. Inoculating plants with these taxa can enhance nitrogen fixation and, in turn, promote plant growth [83]. In our study, we observed that the relative abundance of *Pseudomonas* and the total nitrogen levels increased progressively from the rhizosphere soil to the roots and leaves. Notably, total nitrogen was identified as the most significant factor explaining the variations in the relative abundance of *Pseudomonas*. These findings suggest a strong association between nitrogen availability and the colonization and proliferation of *Pseudomonas* within *S. nigrescens*. The increased nitrogen levels may create a favorable environment for *Pseudomonas*, facilitating their growth and potential nitrogen-fixing activity, which could further contribute to the nitrogen availability within the plant system. This relationship underscores the potential role of *Pseudomonas* in enhancing plant nutrient uptake and growth through nitrogen fixation. In contrast, although total nitrogen is the key factor for variations in their relative abundance, *Sphingomonas* and *Devosia* exhibit a trend opposite to that of nitrogen levels from rhizosphere soil to roots and leaves. This observation implies that the relative abundance trends in *Sphingomonas* and *Devosia* are likely the result of complex interactions between multiple biotic and abiotic factors, rather than being driven solely by nitrogen levels. Further studies incorporating environmental, genetic, and ecological aspects could help unravel these complex interactions. Additionally, we identified total phosphorus as the most significant factor explaining the variations in the relative abundance of *Pseudonocardia* and *Mycobacterium*. Interestingly, the rhizosphere soil exhibited the highest relative abundance of both genera and also had the highest total phosphorus levels. Microbes related to *Pseudonocardia* possess the ability to mobilize organic phosphorus through the secretion of phosphatase enzymes [84]. Research has indicated that *Mycobacterium*-related strains possess genes encoding an inorganic phosphorus transporter system, such as *pstSCAB* [85]. Given these observations, it is evident that total phosphorus levels play a crucial role in affecting the relative abundance of *Pseudonocardia* and *Mycobacterium* associated with *S. nigrescens*.

## 4. Materials and Methods

### 4.1. Site Depiction and Sampling

This study was conducted in the Qiangyong Glacier Basin (28°53′ N, 90°13′ E), situated on the northern flank of the Himalayas (Appendix A). In August 2020, we selected ten healthy individual plants (>20 m apart from each other) at the Qiangyong glacier terminus zone (Appendix A). The glacier terminus zone is positioned between the glacier terminus and the confluence of the west and east glacier runoff. Its formation dates back to the Little Ice Age (approximately 0.13  ±  0.02 to 0.36  ±  0.09 ka BP) [86] (Appendix A). Whole plants with intact root systems were excavated using sterilized spades, shaking off the loosely bound rhizosphere soil. Subsequently, the leaves and roots were collected from each plant, placed in individual sterilized polyethylene bags, labeled, immediately stored in a refrigerator (−20 °C), and transported to the laboratory. Additionally, soil that was not in contact with the root systems and situated at least 50 cm away from each sampled plant was collected after removing the topsoil. All collected soil samples were placed in sterilized plastic bags and immediately frozen at −20 °C until they arrived at the laboratory.

### 4.2. Sample Collection of Rhizosphere Soil, Root Endosphere, and Leaf Endosphere Fractions

The leaves were washed with sterile cooled TE (Tris-EDTA, 10 mM Tris, 1 mM EDTA, pH 7.5) buffer, and then the washed leaves were disinfected via consecutive immersion for 1 min in 80% (vol/vol) ethanol, 5 min in 3.25% (vol/vol) sodium hypochlorite, and 30 s in 80% ethanol. Sterilization was completed with three sequential 2 min rinses in sterile distilled water, followed by leaves that were dried with sterile absorbent paper. To validate the effectiveness of surface sterilization, the sterile water (100 µL) used in the final rinse was added to TSA (Tryptic Soy Agar) and cultured in the dark for 7 d at 28 °C to check for the appearance of colonies. The fibrous roots (diameter < 2 mm) were separated from the taproot using sterilized scissors, and then root samples were placed into 50 mL sterile centrifuge tubes and were washed with PBS buffer (10 mM, pH 7.4) on a shaking table (150 rpm) for 1 h, followed by the separation of fibrous roots from suspension. The soil particles directly dislodged from the fibrous root were defined as the rhizosphere soil, which was pelleted by centrifugation (10,000× *g* for 10 min) in 50 mL sterile centrifuge tubes. The root samples were then separated, surface-sterilized, and verified as described above for leaf samples. In total, 30 samples (1 plant species × 10 individual plants × 3 compartments) were used in this study. All the samples were stored at −80 °C until required for DNA extraction.

### 4.3. DNA Extraction and PCR Amplification

The roots and leaves were aseptically cut, freeze-dried in liquid nitrogen, and then homogenized with a mortar and pestle under aseptic conditions. The genome DNA extracting from homogenized tissues and rhizosphere soils was performed via DNeasy PowerSoil Kit (Mo Bio Laboratories, Carlsbad, CA, USA) with their standard protocol. Qubit double stranded DNA assay kit (Thermo Fisher Scientific, Singapore) was used for quantifying genomic DNA. The extracted DNA was subjected to bacterial 16s region amplification using primers 799F (AACMGGATTAGATACCCKG) and 1193R (ACGTCATCCCCACCTTCC). PCR was performed in a 20 µL reaction solution containing 5 × FastPfu Buffer (4 µL), 2.5 mM of each dNTP (2 µL), 5 µM of each primer (0.8 µL), FastPfu Polymerase (0.4 µL), BSA (0.2 µL), and 10 ng of template DNA. The PCR conditions were set at 95 °C for 3 min, 27 cycles for denaturation at 95 °C for 30 s, annealing at 55 °C for 30 s, and extension at 72 °C for 45 s, followed by a final extension at 72 °C for 10 min. PCR products were cleaned and purified using AxyPrep DNA Gel Extraction Kit (Axygen Biosciences, Union City, California, United States) and quantified using Quantus™ Fluorometer (Promega, Madison, WI, USA). The sequencing library was constructed by the addition of an Illumina sequencing adaptor to the product using NEXTFLEX R Rapid DNA-Seq Kit (Bioo Scientific, Austin, TX, USA), according to the manufacturer’s protocol. Libraries were sequenced on the Illumina MiSeq PE 250 platform with a Paired-End protocol at the Majorbio Bio-Pharm Technology Limited Liability Company (Shanghai, China).

### 4.4. Quantification of Total Nitrogen, Total Phosphorus, and Total Potassium Contents in Soil, Roots, and Leaves

To measure total nitrogen (TN), air-dried soil samples (0.5 g) were digested with a mixture of K_2_SO_4_, CuSO_4_, and Se (100:10:1 ratio) and 5 mL of H_2_SO_4_. After cooling, the digest was diluted to 20 mL with distilled water, filtered, and analyzed colorimetrically at 660 nm using a SEAL AutoAnalyzer 3 (SEAL Analytical, Norderstedt, Germany). The total phosphorus (TP) content in the soil was determined using the NaOH melting-molybdenum antimony colorimetric method. Air-dried soil samples (0.25 g) were fused with NaOH in nickel crucibles at 720 °C using a muffle furnace. The resulting digest was transferred to a 50 mL volumetric flask to fix the solution volume for phosphorus (P) measurement. An aliquot of 2–10 mL of this solution was then mixed with a Mo-Sb chromogenic agent, and the absorbance was measured at 700 nm using a UV-1900i spectrophotometer (Shimadzu, Kyoto, Japan). The total potassium (TK) content was determined by mixing air-dried soil samples (0.25 g) with 2.0 g of NaOH in a nickel crucible and heating at 720 °C for 15 min in a muffle furnace. After cooling, the fusion product was dissolved in 10 mL of deionized water and adjusted to a final volume of 50 mL in a volumetric flask. The emission intensity of the resulting solution was measured using a FP6410 flame photometer (NESA Analytical Instrument Company Limited, Shanghai, China).

The root and leaf samples were first washed with tap water and distilled water, and then oven-dried at 80 °C for 48 h and weighed. The dried samples were ground to pass through a 0.5 mm sieve. A 0.3 g portion of the dry root or leaf sample was soaked in 10 mL sulfuric acid (H_2_SO_4_) for 24 h and subsequently digested in a digestion system within a fume hood, and heated to 180 °C for 3 h, and then 5 mL hydrogen peroxide (H_2_O_2_) was added. The digested solution was transferred into a 100 mL volumetric flask and diluted to 100 mL with deionized water for the analysis of TN, TP, and TK. The TN concentration was analyzed using the SEAL AutoAnalyzer 3 (SEAL Analytical, Norderstedt, Germany). TP concentration was measured using the molybdenum antimony colorimetric method. The TK concentration was determined with a flame photometer (FP6410, China).

### 4.5. Bioinformatic Analysis

Raw sequencing was processed using the bioinformatic pipeline QIIME2 (version 2022.2) [87]. The 16S rRNA gene sequences were demultiplexed using the cutadapt plugin in Qiime2 [41]. The QIIME2 plugin DADA2 was used for quality control, filtering, chimera identification, denoising, the clustering of the sequences to amplicon sequence variation (ASV), and producing the feature table. Sequences were assigned to taxonomy with the QIIME2 plugin feature classifier [88] with pretrained naive Bayes classifiers [89] trained on the SILVA 138 database [90] for the 16S rRNA gene sequences. After then, ASVs that were identified as chloroplast, mitochondria, and unclassified were removed from the 16S rRNA gene sequences. Phylogenetic trees were built using MAFFT alignment in QIIME 2 and the FastTree algorithm [91].

### 4.6. Statistical Analysis

All statistical analyses were conducted in R (version 4.3.3), using the ggplot2 package (version 3.5.1) for visualization. However, the co-occurrence network visualizations were executed using Gephi version 0.10. In addition, the Kruskal–Wallis Test was used to assess significant differences among the groups, followed by Dunn’s test for post hoc multiple comparisons.

The alpha diversity indices, including Richness, Shannon, Pielou’s evenness, and ACE, were calculated using the picante package (version 1.8.2). Niche breadth was assessed using the niche.width function from the spaa package (version 0.2.2). The average variation degree (AVD) was determined according to the approach outlined in [53]. Indicator species analysis of bacterial ASVs was conducted for each plant compartment niche based on ASV relative abundances, using the multipatt function from the indicspecies package (version 1.7.14). ASVs with indicator values > 0.7 and *p* < 0.05 were identified as strong indicators of specific plant compartment niches, reflecting the strength and specificity of their association with plant compartment niches [92]. Three distinct permutation tests were executed: permutational multivariate analysis of variance (ADONIS), analysis of similarity (ANOSIM), and the multiple-response permutation procedure (MRPP). These tests utilized the adonis, anosim, and mrpp functions, respectively, from the vegan package (version 2.6-6.1), and were based on Bray–Curtis and Unifrac dissimilarity measures. For differential abundance analysis, the cpm, glmFit, and glmLRT functions from the edgeR package (version 4.0.16) were used for data filtering, model fitting, and differential expression testing, respectively. SOURCETRACKER (version 1.0), based on a Bayesian approach, was employed to estimate the potential sources [93]. The VennDiagram package (version 1.7.3) was used to determine the shared and unique elements among different groups. The plant compartment niche/bacterium preferences were evaluated based on the methodologies outlined in [9,94].

Co-occurrence networks were constructed by calculating Spearman’s rank coefficients (*r*) between amplicon sequence variants (ASVs). Relationships were deemed statistically robust with *r* > 0.8 and statistically significant with *p* < 0.01. Node topology features, such as node degree and closeness centrality, were computed using the igraph package (version 2.0.3). Additionally, 10,000 Erdős-Rényi model random networks were generated [95]. To further explore keystone species within co-occurrence networks, within-module connectivity (Zi) and inter-module connectivity (Pi) were calculated using the Hmisc package (version 5.1-3) and the igraph package (version 2.0.3). Nodes were classified into four functional types based on their topological characteristics, as described in [96]: Connectors, which exhibit high connectivity between different modules (Zi < 2.5 and Pi > 0.62); Module Hubs, which have high connectivity within a single module (Zi > 2.5 and Pi < 0.62); Network Hubs, which show high overall connectivity both within and between modules (Zi > 2.5 and Pi > 0.62); and Peripherals, which do not display high connectivity in either context (Zi < 2.5 and Pi < 0.62).

The relative significance of the levels and ratios of TN, TP, and TK in explaining variations in alpha diversity and the relative abundance of dominant taxa was analyzed using Aggregated Boosted Tree models [97,98]. To investigate the relationship between the Bray–Curtis community dissimilarity index and the Euclidean distances of TN, TP, TK, TN:TP, TN:TK, TP:TK, and TN:TP:TK values, the Mantel statistic was applied [99]. Furthermore, the relationship between the alpha diversity and average variation degree (AVD) was assessed using Spearman’s correlation.

## 5. Conclusions

Collectively, our findings demonstrate that the bacterial communities associated with *S. nigrescens* vary significantly across the rhizosphere soil, root endosphere, and leaf endosphere. Total nitrogen and total potassium emerged as the crucial factors accounting for the observed differences in diversity and composition, respectively. We further revealed the interconnectedness of bacterial communities among compartment niches of *S. nigrescens* and identified core taxa. Notably, *Pseudomonas* stands out as the predominant taxon among these core bacteria, with total nitrogen being the most significant factor influencing its relative abundance. This study not only advances our understanding of the assembly principles and ecological interactions of the plant microbiome in the sub-nival belt but also provides an integrated perspective on it. Further investigations employing multi-omics technologies—particularly integrated metagenomic and metatranscriptomic approaches, targeting DNA and RNA levels—are required to elucidate the ecological interactions of bacterial communities across distinct niches (e.g., rhizosphere, phyllosphere, and endosphere) in pioneer plants of the sub-nival belt. To accurately reconstruct these microbial ecosystems, a pipeline encompassing comprehensive data collection, rigorous bioinformatic interpretation, methodological benchmarking against gold-standard protocols, and independent experimental validation must be systematically implemented [100].

## Figures and Tables

**Figure 1 plants-14-01190-f001:**
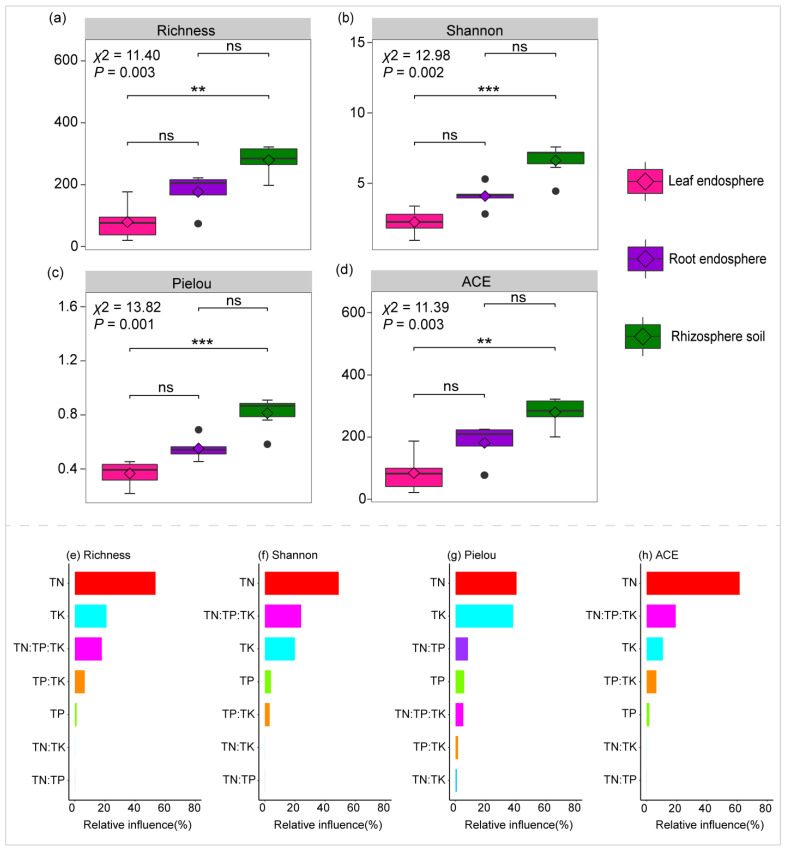
Comparison of bacterial alpha diversity indices, including Richness, Shannon, Pielou, and ACE, across the rhizosphere soil, root endosphere, and leaf endosphere (**a**–**d**). The optimized aggregated boosted tree model identified relationships between bacterial alpha diversity indices and the levels or ratios of TN (total nitrogen), TP (total phosphorus), and TK (total potassium) (**e**–**h**). The Kruskal–Wallis test was used to evaluate significant differences in alpha diversity indices among the rhizosphere soil, root endosphere, and leaf endosphere, followed by Dunn’s test for post hoc multiple comparisons (ns: not significant; ** *p* < 0.01; *** *p* < 0.001).

**Figure 2 plants-14-01190-f002:**
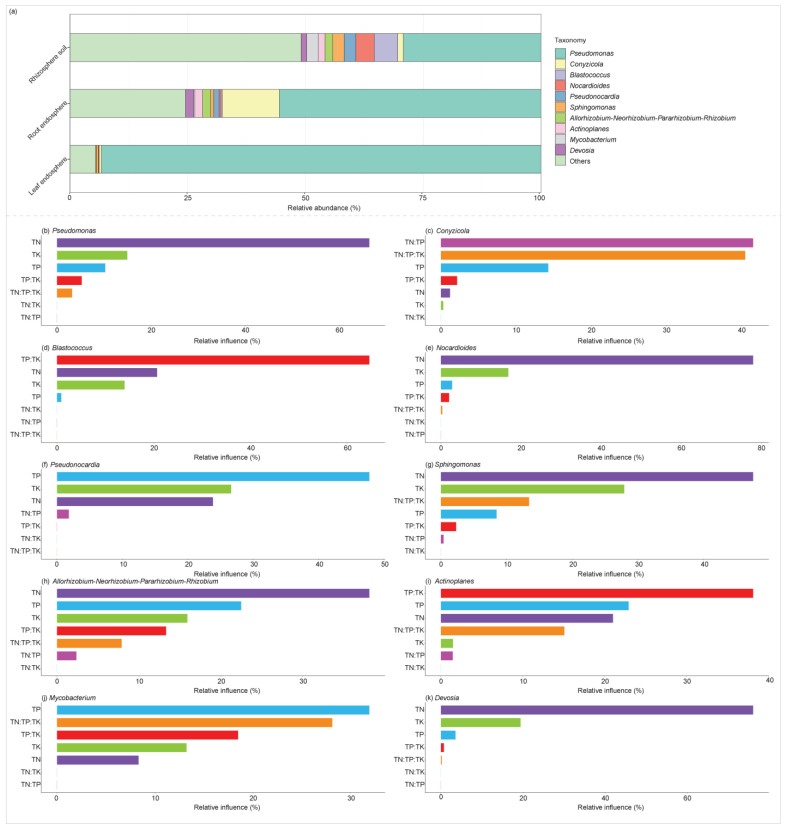
Comparison of the relative abundance of bacterial genera across rhizosphere soil, root endosphere, and leaf endosphere (**a**). Relationships between the levels and ratios of TN (total nitrogen), TP (total phosphorus), and TK (total potassium) and the relative abundance of bacterial genera were identified using an optimized aggregated boosted tree model (**b**–**k**).

**Figure 3 plants-14-01190-f003:**
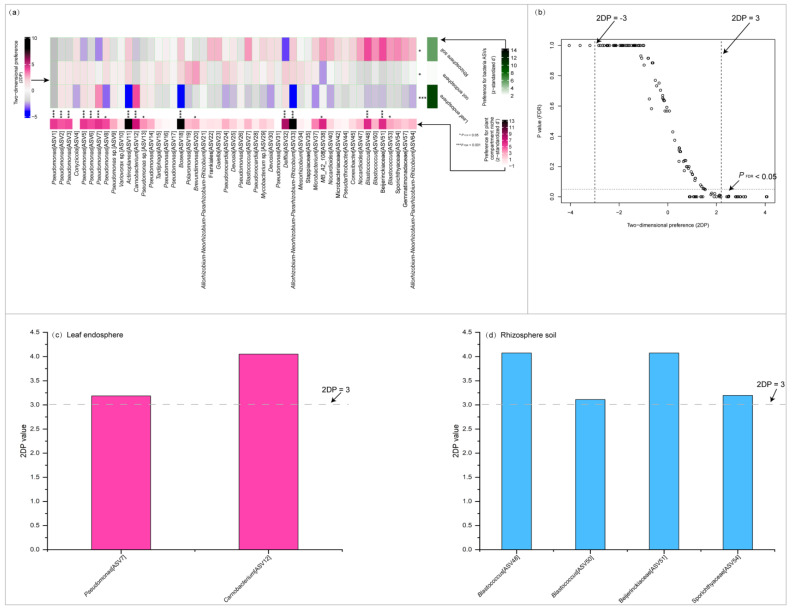
Preferences observed in plant compartment niche–bacteria associations. The standardized d′ estimate of preferences for bacterial amplicon sequence variations (ASVs) is shown for each plant compartment niche (row) (**a**). Likewise, the standardized d′ estimate of preferences for plant compartment niche is indicated for each of the observed bacterial ASVs (column) (**a**). Each cell in the matrix indicates a two-dimensional preference (2DP) estimate, which measures to what extent the association of a plant compartment niche–bacteria pair was observed more or less frequently than would be expected by chance. The *p* values were adjusted based on the false discovery rate (FDR). The relationship between 2DP- and FDR-adjusted *p* values shows that 2DP values larger than 3.0 represented strong preference (PFDR < 0.05) (**b**). The pairs of leaf endosphere (**c**) and rhizosphere soil (**d**) with specific bacterial ASVs exhibited remarkably strong preferences.

**Figure 4 plants-14-01190-f004:**
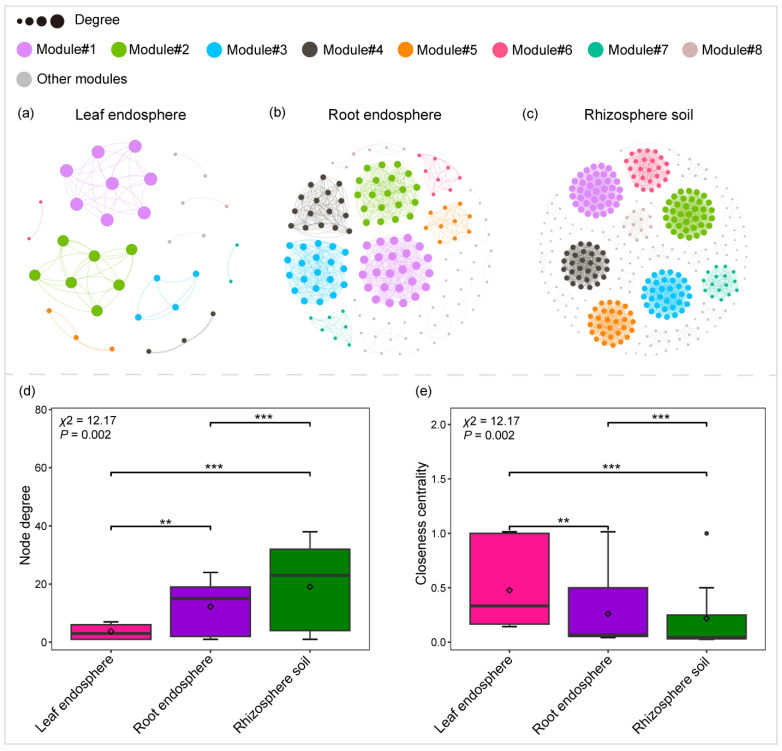
Correlation-based networks of bacterial communities in the rhizosphere soil, root endosphere, and leaf endosphere (**a**–**c**). Bacterial amplicon sequence variants (ASVs) are represented as nodes, while significant statistical correlations between these nodes are illustrated as edges (**a**–**c**). Node size reflects ASV degree, and color indicates the corresponding module (**a**–**c**). Comparison of node-level topological properties across the rhizosphere soil network, root endosphere network, and leaf endosphere network (**d**,**e**). The Kruskal–Wallis test was conducted to assess significant differences in node-level topological properties among the rhizosphere soil, root endosphere, and leaf endosphere, followed by Dunn’s test for post hoc multiple comparisons (** *p* < 0.01; *** *p* < 0.001) (**d**,**e**).

**Figure 5 plants-14-01190-f005:**
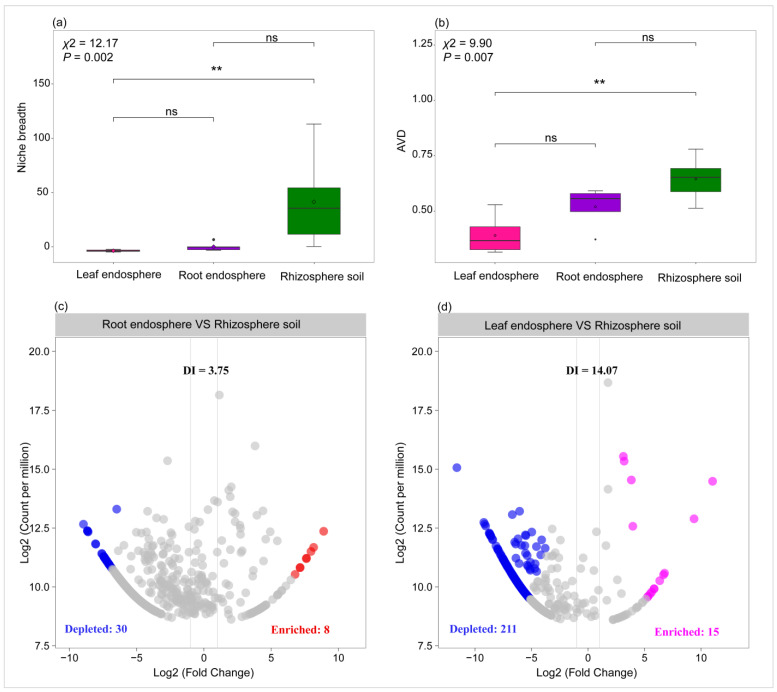
Comparison of niche breadth and average variation degree (AVD) of bacterial community across rhizosphere soil, root endosphere, and leaf endosphere (**a**,**b**). The Kruskal–Wallis test was conducted to assess significant differences in niche breadth and AVD among the rhizosphere soil, root endosphere, and leaf endosphere, followed by Dunn’s test for post hoc multiple comparisons (ns: not significant; ** *p* < 0.01) (**a**,**b**). Volcano plots illustrating the enrichment and depletion patterns of the bacterial community in root endosphere and leaf en-dosphere (**c**,**d**). The Depletion Index (DI), defined as the ratio of depleted ASVs to enriched ASVs, evaluates the magnitude of selective effects exerted by plants on associated bacteria, with higher DI values indicating stronger depletion effects (**c**,**d**) [18].

**Figure 6 plants-14-01190-f006:**
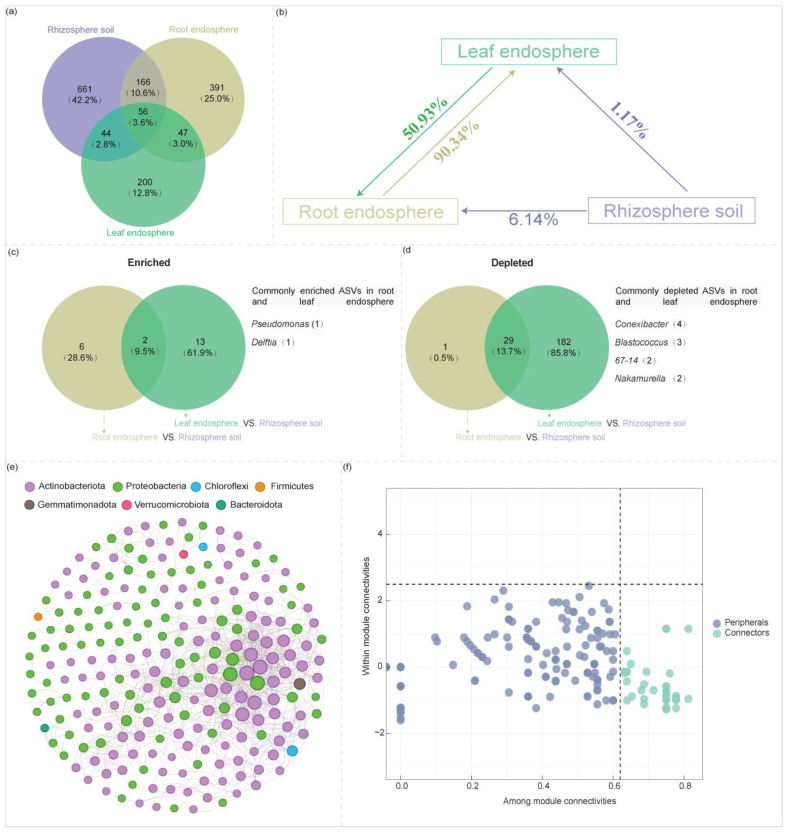
Venn diagram shows the shared and unique amplicon sequence variants (ASVs) among the rhizosphere soil, root endosphere, and leaf endosphere (**a**). Source model shows the potential sources of bacterial communities within the rhizosphere soil, root endosphere, and leaf endosphere, where the directions of the arrows represent the source–sink relationships, and percentages represent the contribution that each source provides (**b**). Venn diagram illustrates the shared and unique bacterial ASVs between the root and leaf endospheres, highlighting both significantly enriched and depleted ASVs (**c**,**d**). Correlation-based networks of bacterial communities, including all taxa in the rhizosphere soil, leaf endosphere, and root endosphere (**e**). ZiPi plot illustrates the distribution of ASVs, encompassing all taxa present in rhizosphere soil, root endosphere, and leaf endosphere, according to their topological roles (**f**).

**Table 1 plants-14-01190-t001:** Comparison of bacterial community composition dissimilarity among the rhizosphere soil (Rs), root endosphere (Re), and leaf endosphere (Le) using three non-parametric statistical methods.

	Multiple Comparison	ADONIS		ANOSIM		MRPP	
		*R^2^*	*p*	*R*	*p*	*δ*	*p*
Bray–Curtis dissimilarity	Le vs. Re vs. Rs	0.453	0.001	0.661	0.011	0.502	0.001
	Le vs. Re	0.307	0.001	0.475	0.004	0.416	0.006
	Le vs. Rs	0.509	0.004	0.850	0.002	0.462	0.005
	Re vs. Rs	0.282	0.004	0.661	0.011	0.631	0.008
Unifrac dissimilarity	Le vs. Re vs. Rs	0.236	0.001	0.532	0.001	0.803	0.001
	Le vs. Re	0.143	0.028	0.245	0.034	0.839	0.021
	Le vs. Rs	0.218	0.001	0.702	0.002	0.804	0.004
	Re vs. Rs	0.195	0.003	0.651	0.003	0.766	0.005

**Table 2 plants-14-01190-t002:** Indicator amplicon sequence variants (ASVs) for leaf endosphere, root endosphere, and rhizosphere soil.

ASV	Taxonomy	Plant Compartment Niche	Indicator Value	*p* Value	Mean Abundance
			(>0.7)	(<0.05)	
ASV1	*Pseudomonas*	Leaf endosphere	0.736	0.003	63.107%
ASV2	*Pseudomonas*	Leaf endosphere	0.886	0.005	8.297%
ASV5	*Pseudomonas*	Leaf endosphere	0.85	0.009	4.248%
ASV7	*Pseudomonas*	Leaf endosphere	0.939	0.003	4.358%
ASV12	*Carnobacterium*	Leaf endosphere	1.000	0.001	1.403%
ASV4	*Conyzicola*	Root endosphere	0.965	0.047	11.944%
ASV19	*Polaromonas*	Root endosphere	0.834	0.023	0.970%
ASV20	*Brevundimonas*	Root endosphere	0.775	0.020	0.967%
ASV32	*Delftia*	Root endosphere	0.805	0.041	0.386%
ASV22	Frankiales	Rhizosphere soil	0.987	0.001	1.492%
ASV23	*Gaiella*	Rhizosphere soil	0.968	0.001	1.345%
ASV27	*Blastococcus*	Rhizosphere soil	0.873	0.003	1.070%
ASV33	*Allorhizobium*-*Neorhizobium*- *Pararhizobium*-*Rhizobium*	Rhizosphere soil	0.82	0.007	0.633%
ASV38	*MB-A2-108*	Rhizosphere soil	0.983	0.001	0.764%
ASV40	*Nocardioides*	Rhizosphere soil	0.964	0.001	0.707%
ASV42	Microbacteriaceae	Rhizosphere soil	0.811	0.024	0.586%
ASV47	*Nocardioides*	Rhizosphere soil	0.970	0.001	0.696%
ASV48	*Blastococcus*	Rhizosphere soil	1.00	0.001	0.754%
ASV50	*Blastococcus*	Rhizosphere soil	0.899	0.003	0.720%
ASV51	Beijerinckiaceae	Rhizosphere soil	1.00	0.001	0.767%
ASV53	*Blastococcus*	Rhizosphere soil	0.816	0.014	0.741%
ASV54	Sporichthyaceae	Rhizosphere soil	0.967	0.001	0.565%
ASV57	Gemmatimonadaceae	Rhizosphere soil	0.939	0.001	0.508%
ASV64	*Allorhizobium*-*Neorhizobium*- *Pararhizobium*-*Rhizobium*	Rhizosphere soil	0.839	0.013	0.461%

**Table 3 plants-14-01190-t003:** Key topological properties of bacterial community co-occurrence networks in the rhizosphere soil, root endosphere, and leaf endosphere.

Network Properties	Leaf Endosphere	Root Endosphere	Rhizosphere Soil
Observed networks			
Edges	66.00	932.00	3424.00
Nodes	35.00	152.00	359.00
Clustering coefficient	1.00	0.99	1.00
Average path length	1.00	1.01	1.00
Modularity	0.71	0.80	0.86
Graph density	0.11	0.08	0.05
Diameter	1.00	1.97	1.00
Average degree	3.77	12.26	19.08
Random networks			
Average path length	0.69 ± 0.09	0.29 ± 0.004	0.29 ± 0.002
Clustering coefficient	0.91 ± 0.03	0.08 ± 0.004	0.05 ± 0.002
Modularity	0.397 ± 0.03	0.23 ± 0.007	0.19 ± 0.004

## Data Availability

The original contributions presented in the study are included in the article; further inquiries can be directed to the corresponding author.

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
