# Peer review of "Differentiation and Interconnection of the Bacterial Community Associated with Silene nigrescens Along the Soil-To-Plant Continuum in the Sub-Nival Belt of the Qiangyong Glacier"

_plants, 2025, doi:10.3390/plants14081190_

Round 1
Reviewer 1 Report
Comments and Suggestions for Authors
The present article have a high quality and solid methodology, with significative results regarding bacterial communities in S. nigrescens. However, it could be improved if the following points could be addressed;
- In the introduction, could you elaborate about why did you choose S. nigrescens
- In the first lines of 2.1 where you describe the number of reads, please add the sample number, i.e. "403,306 raw reads" to "403,306 raw reads across 30 libraries", or similar to provide mo information about the experimental design, as it is not very clear (mainly because journal format requires results before methods).
- On Fig 1, please remove Chao1 metrics from results, it seems redundant with Richness and it is affected by dada2 due the removal of singleton sequences required for Chao1 computation.
- On Fig 2, please redo the figure, Didn't you perform edgeR? why you report Wilcoxon test results, this test is not appropiate for compositional data.
- In Table 2, please add new columns informing the % relative abundance on each compartiment, and clarify the meaning of IndVal value, is this A or B in indicspecies report? maybe it would be better to report both.
- In section 2.3 or Figure 6 caption, please inform what is the meaning for DI, to make it more clear.
Author Response
Dear Reviewer,
Thank you for your valuable feedback. We have carefully revised the manuscript based on your comments. The modifications are highlighted in the revised manuscript, and a point-by-point response to your suggestions is provided in a separate response letter (Answer the comments from the reviewer#1).
Please let us know if further adjustments are needed. We sincerely appreciate your time and effort in reviewing our work.
Yours sincerely,
Wangchen Sonam
Institute of Tibetan Plateau Research, Chinese Academy of Sciences

Reviewer 2 Report
Comments and Suggestions for Authors
The manuscript by Sonam et al., investigated the bacterial community in the Silene nigrescens tissues (root and leaf endospheres) and its rhizosphere soil. The authors did extensive data analysis and showed the bacterial community compositions, diversities, and potential function and interactions of the bacteria in/between the three compartments (root, leaf, rhizosphere soil). I mainly have two questions/comments about this study.
Did the authors measured the abundance of the total bacteria (by e.g., qPCR) in the three compartments? I guess the abundance difference is huge between the three compartments. And so it is not surprise that their bacterial community composition and diversity are significantly different from each other. Another reason I’m asking is because I noticed the bacterial community in the leaf endosphere is almost 100% dominated by Pseudomonas species. So what is the amount these Pseudomonas species? And is such amount sufficient for their function?
Another issue is that the authors only performed 16S rRNA gene based bacterial community analysis. The functional and interactional analyses are pure predictions based on the presence of specific bacterial species, which is extremely unreliable. As such, the significance of this study is limited. Moreover, is bacterial playing a significant role in the microbial ecology of Silene nigrescens tissues and its rhizosphere soil? What about archaea and fungus? Therefore, I would suggest the authors expand this investigation using meta-genomics.
Author Response
Dear Reviewer,
Thank you for your valuable feedback. We have carefully revised the manuscript based on your questions/comments. The modifications are highlighted in the revised manuscript, and a point-by-point response to your questions/comments is provided in a separate response letter (Answer the comments from the reviewer#2).
Please let us know if any further adjustments are needed. We deeply appreciate the time and effort you have devoted to reviewing our work, and we are truly grateful for your invaluable feedback.
Yours sincerely,
Wangchen Sonam
Institute of Tibetan Plateau Research, Chinese Academy of Sciences

Round 2
Reviewer 2 Report
Comments and Suggestions for Authors
The authors improved the manuscript with additional information and clarifications. The earlier comments/questions are addressed fine. Although some questions may not be fully addressed at this time, the manuscript is of value in providing stages for further microbiome analysis.
Regarding the functional and interactional analyses, a lot of papers have been published using 16S rRNA gene sequencing and various bioinformatic tools. However, in-depth data collection and interpretation, and method evaluation and experimental validation are needed to gain insight into the real microbial ecosystem (https://doi.org/10.1038/s41396-019-0459-z).
Another small thing:
The figure in the comment answer (coverletter) file provide some important insight into the function of these Pseudomonas species, and it's better to include this figure in the manuscript.
Author Response
Dear Reviewer,
Thank you for your insightful feedback and recognition of our manuscript's value in advancing periglacial plant microbiome research. We have carefully reviewed your suggestions and provided detailed point-by-point responses. All revisions are clearly highlighted in the manuscript, accompanied by a separate response letter (Answer the comments from the reviewer) addressing each comment individually.
Thank you sincerely for your meticulous evaluation of our work and the thought-provoking suggestions you provided. Your insights have illuminated critical directions for future research on microbial communities associated with periglacial plants. While some of the methodological and conceptual challenges you raised remain beyond the scope of current technical capabilities, they will undoubtedly serve as a guiding framework for our ongoing investigations.
Please let us know if any further adjustments are needed. We deeply appreciate the time and effort you have devoted to reviewing our work, and we are truly grateful for your invaluable feedback.
With heartfelt gratitude,
Wangchen Sonam
Institute of Tibetan Plateau Research, Chinese Academy of Sciences
